# Understanding the Working Mechanism of the Novel HKUST-1@BPS Composite Materials as Stationary Phases for Liquid Chromatography

**DOI:** 10.3390/polym14071373

**Published:** 2022-03-28

**Authors:** Bulat R. Saifutdinov, Vera I. Isaeva, Vladimir V. Chernyshev, Vadim V. Vergun, Gennady I. Kapustin, Yulia P. Ivanova, Mikhail M. Ilyin, Olga P. Tkachenko, Aleksey K. Buryak, Leonid M. Kustov

**Affiliations:** 1A.N. Frumkin Institute of Physical Chemistry and Electrochemistry, Russian Academy of Sciences, Building 4, 31 Leninsky Prospect, 119071 Moscow, Russia; vladimir@struct.chem.msu.ru (V.V.C.); kurnyshevay@mail.ru (Y.P.I.); akburyak@mail.ru (A.K.B.); 2Department of Chemistry, M.V. Lomonosov Moscow State University, Building 3, 1 Leninskie Gory, 119991 Moscow, Russia; lmkustov@mail.ru; 3N.D. Zelinsky Institute of Organic Chemistry, Russian Academy of Sciences, 47 Leninsky Prospect, 119991 Moscow, Russia; polubrat@mail.ru (V.V.V.); gik@ioc.ac.ru (G.I.K.); ot@ioc.ac.ru (O.P.T.); 4A.N. Nesmeyanov Institute of Organoelement Compounds, Russian Academy of Sciences, 28 Vavilov Street, 119991 Moscow, Russia; mil@ineos.ac.ru; 5Institute of Ecotechnology, National University of Science and Technology MISiS, 4 Leninsky Prospect, 119991 Moscow, Russia

**Keywords:** coordination polymers, HKUST-1 metal-organic framework (MOF), biporous silica (BPS), composite materials, adsorption, high-performance liquid chromatography (HPLC)

## Abstract

Composite materials have been used based on coordination polymers or microporous metal-organic frameworks (MOFs) combined with mesoporous matrices for adsorption-related techniques, which enable outflanking some adverse phenomena manifested during pristine components operation and enhance the performance and selectivity of the resulting materials. In this work, for the first time, the novel HKUST-1@BPS composites synthesized by the microwave-assisted (MW) technique starting from microporous HKUST-1 (Cu_3_(btc)_2_) MOF and biporous silica matrix (BPS) with bimodal mesopore size distribution were comparatively studied as materials for liquid-phase adsorption techniques utilizing the high-performance liquid chromatography (HPLC) method and benzene as a model adsorbate. It was established that the studied HKUST-1@BPS composites can function as stationary phases for HPLC, unlike the pristine HKUST-1 and bare BPS materials, due to the synergetic effect of both components based on the preliminary enhanced adsorbate mass transfer throughout the silica mesopores and, subsequently, its penetrating into HKUST-1 micropores. The suggested mechanism involves the initial deactivation of open metal Cu^2+^ sites in the HKUST-1 framework structure by isopropanol molecules upon adding this polar component into the mobile phase in the region of the isopropanol concentration of 0.0 to 0.2 vol.%. Thereafter, at the medium range of varying the isopropanol concentration in the eluent of 0.2 to 0.3 vol.%, there is an expansion of the previously inaccessible adsorption centers in the HKUST-1@BPS composites. Subsequently, while further increasing the isopropanol volume fraction in the eluent in the region of 0.3 to 5.0 vol.%, the observed behavior of the studied chromatographic systems is similar to the quasi-normal-phase HPLC pattern. According to the obtained thermodynamic data, benzene adsorption into HKUST-1 micropores from solutions with a vol.% of isopropanol in the range of 0.4 to 5.0 follows the unique entropy-driven mechanism previously described for the MIL-53(Al) framework. It was found that HKUST-1 loading in the composites and their preparation conditions have pronounced effects on their physicochemical properties and adsorption performance, including the adsorption mechanism.

## 1. Introduction

Over the past four decades, high-performance liquid chromatography (HPLC) has become a prevalent and effective one-of-a-kind tool for the separation and bioanalysis of various natural and synthetic mixtures consisting of thermolabile substances. It is widely used in omics technologies, medical diagnostics, ecology, forensic investigations, petrochemistry, etc. [1,2]. Moreover, in the last 20 years, HPLC has also evolved as one of the key methods for the biophysical and physicochemical characterization of biomolecules and surfaces [2,3]. Currently, most separations by the HPLC technique are carried out primarily utilizing classical stationary phases, such as modified silica gels, graphitized carbons, and styrene-divinylbenzene resins, which, nonetheless, cannot chemically distinguish components of some hard-to-separate multicomponent mixtures. Therefore, it is possible to improve the throughput and selectivity of separations by HPLC using cutting-edge porous materials as stationary phases for chromatographic columns [4,5]. Hence, creating novel sorption materials certainly is an important trend of modern chromatography and simultaneously hyphenated methods, such as liquid chromatography–mass spectrometry. In addition to HPLC, these sorbents can be used in solid-phase extraction and preconcentration coupled with chromatographic and other analytical methods [6,7,8,9,10,11,12]. The stationary phases for HPLC, as is well known, should have a homogeneous surface, constant size and shape of particles, thermal stability, as well as sufficient strength and rigidity. The state-of-the-art class of such materials promising for HPLC is metal-organic coordination polymers, including metal-organic frameworks (MOFs) that exhibit improved properties, the selectivity of which can relatively easily be varied by changing the nature of the metal ion or ligand [11].

To date, MOFs are regarded as the largest class of hybrid nanoporous materials. Their three-dimensional networks are composed of metal ions or small metal-oxide clusters as inorganic nodes connected through coordination bonds with polydentate organic molecule-linkers. MOF matrices show a number of outstanding properties, such as high crystallinity degree, large specific surface area and pore volume, and open pore structure, which is accessible for guest molecules as well as a possibility of the rational design of the framework due to the judicious choice of the inorganic nodes and organic linkers or their post-synthesis modification [13,14,15,16,17,18,19,20,21,22,23]. These physicochemical characteristics make MOFs promising materials for versatile applications in adsorption, separation, and catalysis [24,25,26,27,28]. In terms of liquid-phase adsorption, changing metal ions and organic linkers, as well as their connection modes in a wide range, allows one to obtain metal-organic frameworks with tunable pore size, shape, and functionality, and, therefore, with ingenious performance and selectivity towards different classes of organic compounds simultaneously both in reversed-phase and normal-phase HPLC.

However, many contemporary MOF materials feature insufficient hydrolytic stability. Moreover, the crystallites in the synthesized MOF samples do not exhibit homogeneous size distribution. These circumstances restrict the possibility of the practical applications of the MOF-based adsorbents [29,30,31,32,33]. In particular, as a rule, MOF materials are formed as fine powder composed of polycrystalline particles. It results in some adverse consequences, such as an increase in back-pressure in the chromatographic column, and the possibility of the material subsidence and loss increases simultaneously [34]. In several cases, the liquid-phase separation of organic compounds could be achieved, but the spent analysis time is too prolonged. For instance, the chromatographic column packed with MIL-53(Fe) crystallites showed a high selectivity in the separation of xylene isomers. However, the total elution was achieved in approximately 1 h at 20 °C [35].

A problem of improving the functional properties of MOF matrices can be solved by preparing their high-performance composites with other types of materials, such as metal micro- and nanoparticles, ceramics, polymers, carbons, silica, etc. [34,36,37,38,39,40,41,42,43,44,45]. In particular, the MOF-based composites show a high potential as stationary phases in HPLC columns. For instance, core-shell composites based on MOF layers supported on spherical silica particles have shown a high separation ability and selectivity towards different classes of organic compounds [46,47,48,49,50,51]. The investigation of the sorption ability of UiO-66@SiO_2_ microspheres demonstrated that the retention of aromatic compounds is contributed by π–π-stacking interactions between the analyte molecules and functional groups in UiO-66 organic linkers [47,48,49]. At the same time, the sorption is affected by the sizes of both eluted molecules and UiO-66 pores. Therefore, the molecular sieving UiO-66 performance could be revealed [50].

Moreover, the separation of xylene isomers on the pristine HKUST-1 matrix failed. On the contrary, using its composite with silica microspheres with surface carboxylic groups results in good separation of the studied analytes [36,52]. An effective separation of styrene and benzene was achieved on the sorbents obtained by the synthesis of the HKUST-1 crystallites inside pores of monodispersed spheres of silica [53]. Note, the performance of this composite was more efficient in many ways than using individual composite components. First, by elution of the mixtures of analytes through the column packed with the HKUST-1 sorbent, a high back-pressure at the inlet along with chromatographic peak broadening were observed due to a wide particle size distribution. On the contrary, the homogeneous particles of pure Nucleosil demonstrated a low back-pressure in the column under elution conditions, while chromatographic peaks for ethylbenzene and styrene were narrow for individual compounds but inseparable for their mixture. Therefore, the separation of aromatic compounds was implemented due to interactions of their π-electron system with open Cu^2+^ sites. Additionally, a spherical form of silica particles contributed to elution time shortening, improvement in the separation of the chromatographic peaks, and back-pressure decrease at the column inlet.

It was demonstrated [54,55] that preparing the composites based on MOF matrices allows one to eliminate some drawbacks accompanied by the exploitation of their individual components. For instance, the deposition of MIL-101(Fe) crystallites on the mesoporous silica nanospheres (MSN) prevents their aggregation, and thereby eliminates the decrease in the adsorption characteristics of MSN due to particle size enlargement and increase in the resistance to mass transfer. Additionally, MIL-101(Fe) supporting on the MSN particles with diameters of approximately 30 nm contributes to the magnification of the specific surface area of the resulted composite.

The MSN@MIL-101(Fe) composite obtained by a solvothermal method was used as an adsorbent for the extraction and concentration of phytohormones before the analysis of their solution by the HPLC method. It demonstrated a high efficiency of the extraction of the studied components due to strengthening of electrostatic, hydrophobic, π–π-interactions, and hydrogen bonding [55].

Recently [56], we have studied the peculiarities of the mechanism of the liquid-phase adsorption of aromatic compounds on the HKUST-1 material. The microporous HKUST-1 (Cu_3_(btc)_2_, btc = benzene-1,3,5-tricarboxylate) matrix is one of the best studied MOF materials. It has a high specific surface area (BET, up to 1800 m^2^/g) and porosity combined with open metal sites (Cu^2+^) as inorganic nodes in the framework, which are potential adsorption centers [57].

It was found that the selective separation of analyte molecules on these sorbents was determined by π–π-stacking interactions with benzene-1,3,3-tricarboxylate linkers in the HKUST-1 framework along with a possibility of the competitive adsorption of polar solvent molecules on open adsorption Cu^2+^ sites. The role of steric hindrances for the transport of organic molecules in the narrow framework pores was also established. For instance, the retention of diphenylamine and paracetamol molecules from acetonitrile solution was higher than that of tropic acid and ibuprofen because the former molecule has a more planar geometry than the latter one. It was shown that the benzene retention factor increased with a decrease in the mobile phase flow rate. It means that there is no adsorption equilibrium at the middle flow rates of the mobile phase because the space of the HKUST-1 micropores is probably accessible for the adsorbate molecules only at low flow rates of the mobile phase due to the absence of diffusion limitations. Therefore, the further enhancement of the retention ability and adsorption performance for the HKUST-1 based materials through the development of their composites remains highly topical.

In this context, mesoporous silica matrices (MMSs) show a significant potential for the improvement of the sorption characteristics of the MOF materials. Their particles show a high hydrolytic and mechanical stability [58]. MMSs have a high specific surface area and appropriate pore size that assists the retention of extremely bulky molecules of physiologically active compounds, thereby providing their practical applications in liquid-phase adsorption and drug delivery systems in living organisms. A high density of silanol groups on the MMS surface assists the further functionalization, including the preparation of composites with MOF materials. Therefore, in MOF@MMS composites, the synergetic adsorption properties as a result of a number of cooperation effects between intrinsic physicochemical characteristics and adsorption properties of two types of materials can be realized. In particular, creating composite materials based on MMS and HKUST-1 is important because, according to the previous considerations, they, as stationary phases under conditions of HPLC, can reach adsorption equilibrium at middle flow rates of the mobile phase due to the presence of transport mesopores in the silicate matrix.

For instance, the ZIF-8@SBA-15 composite demonstrated a synergetic effect for an increase in the adsorption ability in the removal of bisphenol A from water solutions [12]. Moreover, π–π-stacking interactions and hydrogen bonding between bisphenol A molecules and ZIF-8 crystallites contribute primarily in the adsorption mechanism. In its turn, the MMS component accelerates the sorption process and provides a possibility of easy regeneration and repeated utilization of the ZIF-8@SBA-15 composite.

Among other MMSs, biporous silicas (BPSs) with a bimodal pore size distribution take a specific place. The two kinds of mesopores in the BPS matrices assist the efficient selective adsorption of molecules of different sizes, including biomolecules such as hemoglobin [59].

There are works in which the high-performance and selective liquid-phase separation and deep purification of organic substances on MOFs and their composites were carried out [12,36,52]. However, insights on adsorption mechanisms are frequently controversial and groundless because a scrutiny of the physicochemical fundamentals of liquid-phase adsorption onto MOFs-based materials upon HPLC and solid-phase extraction is still absent. Therefore, it is necessary to investigate the surface chemistry of composite materials based on MOFs utilizing the HPLC technique to understand the working mechanism of these stationary phases and find optimum conditions for their operation.

This work aims to comprehend and establish the working mechanism of the novel HKUST-1@BPS composites as stationary phases under HPLC conditions compared to the pristine HKUST-1 material and the bare BPS silica with bimodal mesopore size distribution. Instead of the previously performed and well-described studies of the retention mechanisms utilizing model compounds of different chemical structures, our efforts were intended to investigate in detail the fundamental relationships between the retention factor and the chromatographic conditions using benzene as a model adsorbate. Particular attention was also paid to investigating the impact of the HKUST-1 content in the composites on their adsorption performance, including an implemented adsorption mechanism. To study the effect of the preparation conditions on the adsorption properties of the HKUST-1@BPS composites, they were synthesized utilizing a microwave-technique following different original procedures.

## 2. Materials and Methods

All reagents and solvents employed were commercial products (Acros Organics, Antwerpen, Belgium). N,N-dimethylformamide (DMF) was distilled over CaH_2_ under a reduced pressure. The biporous silica matrix (BPS) has been synthesized in the Prof. I. I. Ivanova laboratory (Moscow State University) [60].

### 2.1. Preparation of the HKUST-1@BPS Composite Materials and HKUST-1 Reference Sample

All HKUST-1@BPS composite materials and the HKUST-1 reference sample were prepared under microwave (MW) activation of the reaction mass at atmospheric pressure according to original procedures. Elemental analysis was carried out by total X-ray reflection fluorescence analysis using an S2 PICOFOX spectrometer (Bruker, AXS, Munich, Germany).

#### 2.1.1. HKUST-1 *mw* Reference Sample

Cu(NO_3_)_2_ 3H_2_O (2.077 g, 8.58 mmol) was dissolved in H_2_O, while H_3_btc (1.000 g, 4.76 mmol) was dissolved in absolute DMF (20 mL). Both solutions were stirred for 20 min using a magnetic stirrer; then, the mixture was transferred into a glass ampoule and heated at an atmospheric pressure in a chamber of a Vigor MW oven (200 W, 30 min, 125 °C; Vigor, Moscow, Russia). The blue product was rinsed with DMF (4 × 15 mL) and CHCl_3_ (2 × 25 mL), then dried under a vacuum (150 °C, 12 h, 10^–^^2^ Torr, 180 °C, 4 h, 10^–^^2^ Torr).

#### 2.1.2. HKUST-1@BPS Composites

**1**. The mixture of H_3_btc (1.000 g, 4.76 mmol), Cu(NO_3_)_2_·3H_2_O (2.077 g, 8.58 mmol) and BPS, 3.000 g) was ground in an agate mortar, then transferred into a glass ampoule containing H_2_O–DMF (vol. ratio was equal to 1:1) and heated at an atmospheric pressure in a chamber of a Vigor MW oven (200 W, 30 min, 104 °C; Vigor, Moscow, Russia). The light-blue product was rinsed with DMF (4 × 15 mL) and CHCl_3_ (2 × 25 mL) and subsequently dried under a vacuum (150 °C, 12 h, 10^–^^2^ Torr, 180 °C, 4 h, 10^–^^2^ Torr).

**2**. The mixture of H_3_btc (1.000 g, 4.76 mmol), Cu(NO_3_)_2_ × 3H_2_O (2.077 g, 8.58 mmol) and BPS, 2.000 g) was ground in an agate mortar, then transferred into a glass ampoule containing H_2_O–DMF (vol. ratio = 1:1) and heated at an atmospheric pressure in a chamber of a Vigor MW oven (200 W, 30 min, 104 °C; Vigor, Moscow, Russia). The light-blue product was rinsed with DMF (4 × 15 mL) and CHCl_3_ (2 × 25 mL), then dried under a vacuum (150 °C, 12 h, 10^–^^2^ Torr, 180 °C, 4 h, 10^–^^2^ Torr).

**3**. H_3_btc (1.000 g, 4.76 mmol) was dissolved in absolute DMF (15 mL); Cu(NO_3_)_2_ × 3H_2_O (2.077 g, 8.58 mmol) was dissolved in H_2_O (20 mL). Subsequently, both solutions were combined and transferred into a glass ampoule containing BPS ground in an agate mortar (6.000 g). The reagents mixture was heated at an atmospheric pressure in a chamber of a Vigor MW oven (200 W, 30 min, 104 °C; Vigor, Moscow, Russia). The light-blue product was rinsed with DMF (4 × 15 mL) and CHCl_3_ (2 × 25 mL), then dried under a vacuum (150 °C, 12 h, 10^–^^2^ Torr, 180 °C, 4 h, 10^–^^2^ Torr).

### 2.2. Characterization of the Composite Materials and HKUST-1 Reference Sample

#### 2.2.1. N_2_ Adsorption Data

These data were obtained at 77 K using an ASAP 2020 Plus instrument (Micromeritics, Norcross, GA, USA). The specific surface areas were calculated according to the BET equation (ISO 9277). The total pore volume was evaluated at p/p^0^ = 0.99. The distribution in sizes of mesopores was calculated from the desorption branch according to the method of Barrett, Joyner, and Halenda (BJH). The cumulative volume at desorption in the BJH method at 2 nm was taken as the mesopore volume. The thickness of the adsorbed multilayer on the mesopore surface was taken into account. The micropore volume was calculated as the difference between the total pore volume and the mesopore volume. The micropore size distribution was calculated according to the Horwath–Kawazoe model in assumption of a cylindrical shape of the pores [61].

To probe the mechanical robustness of the HKUST-1@BPS composite material (sample **1**) needed for the operation under HPLC conditions, a portion of this activated sample was transferred in a molding form and pressed at 50 atm (air) using a hydraulic pellet press (Perkin–Elmer, New Brunswick, NG, USA) maintaining this pressure during 2 min. The pressed **1** sample was crashed into grains, and the N_2_-low temperature adsorption was measured for them.

#### 2.2.2. Electron Microscopy Study

Target-oriented approach was utilized for the optimization of the analytic measurements [62]. Before measurements, the samples were mounted on a 25 mm aluminum specimen stub and fixed by conductive graphite adhesive tape. Samples morphology was studied under native conditions to exclude metal coating surface effects. The observations were carried out using Hitachi SU8000 field-emission scanning electron microscope (FE-SEM). Images were acquired in secondary electron mode at 2 kV accelerating voltage and at working distance 8–10 mm. Sample morphology were studied using a SU8000 field-emission scanning electron microscope (FE-SEM, Hitachi, Mannheim, Germany).

Before TEM measurements, the samples were mounted on a 3 mm copper grid with lacey carbon film and fixed in a grid holder. Samples morphology was studied using Hitachi HT7700 transmission electron microscope (TEM). Images were acquired in bright-field TEM mode at 100 kV accelerating voltage.

#### 2.2.3. Powder X-ray Diffraction

X-ray powder diffraction data were collected in a reflection mode using an EMPYREAN instrument (PANalytical, Malvern, UK) equipped with a linear X’celerator detector and non-monochromated Cu Kα radiation (α = 1.5418 Å), measurement parameters: tube voltage/current 40 kV/35 mA, divergence slits of 1/8 and 1/4°, 2θ range 5–40°, speed 0.2° min^−1^.

#### 2.2.4. DRIFTS Examinations

Diffuse-reflectance infrared Fourier-transform spectra (DRIFTS) were recorded at room temperature with a Nicolet 460 Protégé spectrometer equipped with a diffuse-reflectance attachment. The samples were placed in the ampoules with a KBr window. The CaF_2_ powder was used as a standard. For a satisfactory signal-to-noise ratio, 500 spectra were collected. The spectra were measured in the wavenumber range of 6000–750 cm^−1^ in 4 cm^−1^ steps.

### 2.3. Chromatographic Study of the HKUST-1@BPS Composite Materials and the Pristine HKUST-1 and BPS Reference Samples

Before HPLC columns were manufactured, the samples were evacuated at 20 °C (3 h) and 100 °C (2 h) to remove physisorbed gases and water.

#### 2.3.1. HPLC Columns Manufacture and Determination of Their Hold-Up Volumes

The HKUST-1@BPS composites and the pristine HKUST-1 and BPS reference samples powders as suspensions in chloroform were packed in the steel chromatographic columns by a pneumatic pump (Alltech, Weingarten, Germany). This pump was connected to a special cylindrical tank for a suspension with a volume of 25 mL. When packing the column, using a manometer, we ensured that the working pressure in the system did not exceed 200 atm. Before filling into the columns, the suspensions of the HKUST-1@BPS, HKUST-1, and BPS powders in chloroform were decanted. Moreover, we filtered off coarse particles on a copper mesh and treated this suspension with an ultrasonic rod (1–2 min). Chloroform was used as a solvent when filling a suspension into the column. The dimensions of the columns filled with granules of the HKUST-1@BPS composites and the pristine HKUST-1 and BPS reference samples were 50 × 4 mm (**1**), 75 × 3 mm (**2**), 50 × 3 mm (**3**), 50 × 4.6 mm (pristine HKUST-1), 70 × 3 mm (bare BPS).

The values of the column hold-up volume *V*_M_ (μL) were determined utilizing static pycnometry, which consists in weighing the masses of the chromatographic column filled with two distinct solvents of different densities [63]. Chloroform, isopropanol, and *n*-hexane were used to determine the hold-up volumes of the chromatographic columns filled with the composite materials and the pristine HKUST-1 and BPS powders. The values of the volume of a stationary phase *V*_S_ (μL) were calculated as a difference between the column geometrical volume *V*_C_ (μL) and the hold-up volume *V*_M_ (μL). The values of phase ratio *φ* were estimated by the following Formula (1) [2]:*φ* = *V*_S_/*V*_M_.(1)

#### 2.3.2. Investigation of the Influence of the Mobile Phase Flow Rate on the Benzene Retention upon HPLC Using the HKUST-1@BPS Composite Materials

To investigate the working mechanism of the examined composites, benzene has been chosen as a test adsorbate.

An Agilent 1200 Series liquid chromatograph (Agilent Technologies, Waldbronn, Germany) equipped with a diode array detector, a Rheodyne dispensing valve (Waters, Milford, MA, USA) with a loop (20 μL), and thermostated columns was used for chromatographic measurements.

To study the accessibility of micropores of the HKUST-1@BPS materials under conditions of HPLC towards molecules of an adsorbate, the benzene retention dependences on the mobile phase flow rate were obtained. The tests were carried out at the constant eluent composition (99.5% of *n*-hexane and 0.5% of isopropanol) and temperature 35 °C. The retention times of benzene were measured at different eluent flow rates: 50, 100, 200, 300, 400, and 500 µL/min. The volume of the probe injected into the chromatographic system was 20 µL. We used the extremely dilute benzene solutions in the examinations of the liquid-phase adsorption mechanism. The volume fraction of benzene in a solution was 10^−3^. Thus, the liquid-phase adsorption was studied under conditions of extremely low degrees of the coverage of the adsorbent surface (the so-called Henry range). The column’s back-pressure in these tests was in the range of 5–59 bar.

Output curves (chromatograms) were detected at a wavelength of the detector λ = 254 nm. The shapes of the output curves measured in the HPLC experiments were close to the correct Gaussian peak shape. An example of the output curve recorded during the elution of benzene at 65 °C throughout the porous HKUST-1@BPS (sample **1**) layer is presented in Appendix A. The benzene retention time *t*_R_ (min) was determined on a maximum of the peak of the output curve. Further, the values of the retention factor (*k*) were calculated by the following well-known Formula (2) [2]:*k* = (*V*_R_ − *V*_M_ − *V*_EC_)/*V*_M_,(2)
where *V*_R_ = *t*_R_·*F* is the benzene retention volume, μL, *V*_EC_ is the extra-column volume in the applied chromatographic system, μL.

#### 2.3.3. Study of the Impact of the Mobile Phase Composition on the Benzene Retention upon HPLC Using the HKUST-1@BPS Composite Materials

To understand intermolecular interactions between adsorbate molecules and active adsorption sites in the HKUST-1@BPS pores while changing the bulk liquid phase composition, the dependences of the retention factor on the composition of a mobile phase were measured. The impact of the mobile phase composition on the benzene retention on the HKUST-1@BPS adsorbents and the pristine HKUST-1 and BPS materials was studied as follows. Before HPLC measurements, the columns packed with HKUST-1@BPS, HKUST-1, and BPS particles were thermostated at 35 °C and rinsed with a mobile phase of different compositions and a volumetric flow rate *F* = 50 μL/min for 40 min. The retention times for benzene were obtained under these constant conditions and at the following volumetric ratios of *n*-hexane to isopropanol in the eluent: 100/0, 99.9/0.1, 99.8/0.2, 99.7/0.3, 99.6/0.4, 99.5/0.5, 99/1, and 95/5% (vol.). The column’s back-pressure in these tests was around 5 bar.

#### 2.3.4. Examination of the Effect of the Chromatographic Column Temperature on the Benzene Liquid-Phase Adsorption onto the HKUST-1@BPS Composite Materials

To figure out the energetics of the liquid-phase adsorption, to determine the temperature dependence of retention, and to calculate the thermodynamic characteristics of benzene adsorption onto HKUST-1@BPS samples, we measured the dependences of the benzene retention times on the column temperature during adsorption under HPLC conditions. The column temperature was changed in the range of 35–65 °C with a step of 10 °C under other equal conditions. The flow rate of the mobile phase was 50 µL/min. The thermodynamics of benzene adsorption was studied from pure *n*-hexane as well as a *n*-hexane–isopropanol binary mixtures with the following volumetric ratios of *n*-hexane to isopropanol: 100/0, 99.9/0.1, 99.8/0.2, 99.7/0.3, 99.6/0.4, 99.5/0.5, 99/1, and 95/5% (vol.). When changing the temperature before the probe injection, each column was thermostated for 20 min for the achievement of the equilibrium. The column’s back-pressure in these tests was not more than 5 bar.

Using the retention factor, the values of the distribution constant (*K*_c_), Gibbs energy (∆*G*°), enthalpy (∆*H*°), and entropy (∆*S*°) of the liquid-phase adsorption upon HPLC were calculated by the following well-known Formulas (3)–(5) [64]:*K*_c_ = *kV*_M_/*V*_S_ + 1,(3)
∆*G*° *=* −*RTlnK*_c_,(4)
ln*K*_c_ = −∆*H*°/*RT* + ∆*S*°/*R*,(5)
where *V*_S_ is the volume of the adsorbent (stationary phase) in the chromatographic column, μL; *R* is the universal gas constant, J/(mol K); *T* is the absolute thermodynamic temperature, K.

The distribution constant reflects the affinity between an adsorbate molecule and an adsorbent. The values of the standard molar differential Gibbs energy, enthalpy, and entropy of the liquid-phase adsorption calculated by this method reflect changes in the energy and number of degrees of freedom of the thermodynamic system during the transition of 1 mole of benzene molecules from the bulk solution into the adsorption phase. The error in calculating the thermodynamic characteristics of adsorption was not higher than 10%.

## 3. Results and Discussion

### 3.1. Synthesis of the Composite Materials

The HKUST-1@BPS composites were synthesized by original procedures according to an in situ approach, which involves adding an appropriate amount of BPS silica in the reagent mixture for the HKUST-1 synthesis. For the composite preparation, MW-activation of the reaction mass at an atmospheric pressure was used. In order to evaluate a direct impact of the synthesis procedures on the physicochemical characteristics of the produced composites, their preparation was carried out in parallel experiments. The differences in the synthesis mode used for producing samples **1**, **2,** and **3** are related with the preliminary mixing of reagents and BPS ground together followed by adding the solvent in the case of composites **1** and **2**, while the material of **3** was prepared starting from combining the separate solutions of the Cu^2+^ source and H_3_btc followed by adding the combined solution to the BPS silica. It could be suggested that, in the case of reagent grinding, the mechanochemical reaction takes place before MW-heating. Another difference is related to the different quantities of BPS silica at the same amount of the Cu^2+^ source and H_3_btc introduced in the composite synthesis.

### 3.2. Characterization of the Composite Materials

The comparison of the SEM micrographs of the BPS silica (Figure 1a), the HKUST-1 reference sample (Figure 1b), and the novel composite **1** demonstrates that the morphology of the latter (Figure 1c) differs from both components; however, its morphology is closer to that of BPS silica. In particular, this material is composed of BPS particles with a fraction of HKUST-1 crystals with sizes of 500 to 1500 nm. Such a size is significantly smaller than the crystal size of the HKUST-1 reference sample (approximately 20–30 μm), also synthesized under MW-activation. This decrease in the HKUST-1 particle size can be related to the surface effect of BPS. A similar surface restrictive effect for the MOF crystals is observed for other types of composites, for instance, for the membranes with a supported polycrystalline layer [65].

The TEM study (Figure 2) revealed the difference in the microstructure of the HKUST-1@BPS composites synthesized in the MW-fields and dispersion of the HKUST-1 particles located in them. Sample **1,** synthesized by grinding together all the reagents and using BPS loading of 3.0 g, has relatively small HKUST-1 nanocrystallites, with sizes of approximately 10 to 20 nm embedded mainly inside the BPS matrix and some bigger HKUST-1 particles with sizes of 500 to 1000 nm located on the surface (Figure 2b).

In the case of composite **2,** synthesized according to the same approach by using a smaller BPS loading (2.0 g), it has HKUST-1 crystallites with rather similar sizes to composite **1**. There are rather big particles with sizes of approximately 500 to 1500 nm on the composite surface (Figure 2c). However, there is a fraction of small HKUST-1 nanocrystals with sizes of approximately 20 to 30 nm embedded in the BPS matrix. Therefore, changing the BPS loading in the composite system does not remarkably affect the HKUST-1 dispersion.

Using another synthesis technique (sample **3**) based on combining separate reagents solutions (without their preliminary grinding) and ground BPS silica taken in a larger amount (6.0 g) than in the cases of composites **1** and **2** resulted in the formation of two fractions of HKUST-1 crystallites in the composite (Figure 2d). The first one is composed of particles with sizes around 500 nm located on the surface. Probably, this rather moderate size of the crystallites located on the BPS surface may be related to the biggest BPS quantity involved in the composite **3** synthesis. The second fraction consists of crystallites with sizes of approximately 50 to 100 nm embedded in the BPS matrix. Note, this composite has the largest HKUST-1 crystallites inside the BPS matrix as compared to **1** and **2** materials. However, its particle size is still much smaller than in the HKUST-1 reference sample.

Thus, the preparation technique involving preliminary reagents grinding, including silica, along with an appropriate silica amount, plays a decisive role in a high dispersion of HKUST-1 crystallites and their homogeneous distribution on the surface and inside the composite.

### 3.3. Structural Examinations

The peak positions in the powder patterns of the three HKUST-1@BPS samples—**1**, **2**, and **3**—are close (Figure 2) and correspond to the cubic unit cell in the space group Fm-3m. The Pawley fitting [66] (see also Appendix A in ESI) performed with the program MRIA [67] confirmed this observation, leading to the following values of the cubic unit cell parameters: a = 26.270(2), 26.392(2), and 26.397(2) Å for **1**, **2,** and **3**, respectively.

Although the cubic HKUST-1 framework in all three samples remains unchanged, the appearance of a wide amorphous halo (in the 2θ region 15–30°) in the powder patterns of **2** and **3** clearly indicates the increase in the content of the amorphous component in these two samples. At the same time, the essential difference in the heights of low-angle peaks and the increase in the cubic unit cell parameter *a* in **2** and **3** as compared with **1** reflect some changes in the crystal structure. Most probably, these structural changes are caused by changes in the pore size distribution. For instance, the intensity of the Bragg peak at 2θ~5.8° (*hkl* = *111*) depends on the content of the pores; in the idealized case of empty pores, its intensity is near zero (see powder patterns 2 and 3 in Figure 3). However, when the pores are filled with the scatterers, the intensity of this peak (*hkl* = *111*, see powder pattern 1 in Figure 3) is directly dependent on the quantity of the electrons in the pores.

### 3.4. Textural Characteristics of the HKUST-1@BPS Composite Materials

The textural properties of the HKUST-1@BPS composites with a different HKUST-1 loading, HKUST-1 reference sample, and BPS matrix calculated from N_2_-adsorption isotherms (Figure 4) are listed in Table 1. The pure HKUST-1 material has the highest specific surface area (BET) as compared to the composites and the bare BPS matrix, which shows the lowest specific surface area value. The surface area values calculated for the HKUST-1@BPS samples from N_2_-adsorption measurements are rather close and have a tendency to increase along with the HKUST-1 content in the composite afforded from the BPS silica introduced in the synthesis (Table 1). Accordingly, the **2** system with the highest HKUST-1 loading features the highest surface area as compared to the other composites, and the **3** material containing a lower content of the HKUST-1 component than its **1** and **2** counterparts has the lowest specific surface area.

It can be seen from the pore volume values of the HKUST-1 ***mw*** reference sample, bare BPS, and HKUST-1@BPS composites (Table 1) that the HKUST-1 crystallites introduce the microporosity in the novel composites. Actually, the micropore content in the hybrids increases along with HKUST-1 loading.

On the other hand, the different HKUST-1 loading determined by the HKUST-1: BPS ratio used in the synthesis results also in the different mesopore content in the composites. Therefore, the **3** material has more mesopores than the **1** and **2** systems. This phenomenon could be explained by a partial mesopore blockage with HKUST-1 crystallites. A similar effect was observed in Ref. [57] related to the HKUST-1@BPS produced by the solvothermal method. In the cited work, the mesoporosity drop was remarkably more pronounced than in this study. It could be suggested that the solvothermal synthesis results in the dominating location of HKUST-1 crystallites inside the silica matrix, and, in the case of the composites synthesized by the MW-technique, many more HKUST-1 particles are situated on the BPS surface.

In particular, the % HKUST-1 content in the composites (Table 1) could be calculated as V_micro_/V_micro HKUST-1_, taking into account that the HKUST-1 reference sample has almost no mesopores, while the BPS matrix has no micropores. The values of the HKUST-1 content in the composites calculated from adsorption measurements are in good consistency with the values calculated from elemental analysis data.

To study the mechanical strength of the HKUST-1@BPS composite (sample **1)** for the evaluation of its potential application as a stationary phase in HPLC, this material has been pressed at 50 atm. The comparison of the textural characteristics for intact and pressed composites (Table 1) shows that the **1** material before pressing has a bit higher specific surface area and significantly higher mesopore volume than its pressed counterpart. On the contrary, the pressing increases the micropore volume for the **1** composite, probably due to degradation of mesopores.

The N_2_-low temperature adsorption isotherms measured for the HKUST-1@BPS composites and the bare BPS silica (Figure 4) are similar and belong to the IV type characteristic for the mesoporous solids. There is a well-defined hysteresis loop in the region of 0.7 to 1.0 of the relative pressure (p/p^0^) on these isotherms. On the contrary, the HKUST-1 reference sample shows a type I isotherm characteristic for microporous materials.

The composite materials and BPS silica feature the similar bimodal mesopore size distribution (Figure 5). The larger fraction is represented by small mesopores with a rather narrow distribution in diameters, i.e., around 2 to 3 nm. The second one includes larger mesopores, with sizes changing in a broad region, i.e., 7 to 40 nm, with an average diameter around 18 nm. The large pore diameter could represent the intercrystalline distances between the primary SiO_2_ particles [60].

These results clearly show the determining role of the BPS component in the formation of the mesoporous fraction of the synthesized composites.

It can be seen from Figure 6 that the HKUST-1 reference sample has micropores with a narrow size distribution in the 0.6 to 0.9 nm region. It is in a good accordance with the micropore diameter reported for the pristine HKUST-1 metal-organic framework [56].

### 3.5. DRIFTS Examinations of the HKUST-1@BPS Composite Materials

The chemistry of the pore surface of the synthesized HKUST-1@BPS composites was studied by the DRIFTS method.

Appendix A shows DRIFT spectra in a wide range recorded for the **1**, **2**, and **3** composite materials and pristine HKUST-1 ***mw*** sample.

Figure 7 and Figure 8 compare the spectra of these samples in two frequency ranges in which absorption bands appear.

The broad band at 3600 to 3000 cm^−1^ in the spectra of the **1** and **2** composites corresponds to the stretching vibration of OH groups (Figure 7) [68,69,70,71,72,73]. The absence of bands in the spectra around 5200 cm^−^^1^ (Appendix A) makes it possible to exclude the presence of adsorbed water not only in the parent HKUST-1 ***mw*** material but also in the composite samples [74]. In the same region of the IR spectra, there may be bands from stretching vibrations of the N–H bond, indicating the presence of the residual solvent (DMF) [74,75].

It can be seen in Figure 7 that the spectra of all the composite materials somewhat differ from each other and from the parent HKUST-1 ***mw*** material. In the spectrum of the **1** sample, four bands at 3091, 2933, 2881, and 2811 cm^−1^ are observed in the entire range; in the spectrum of sample **2**, five bands at 3082, 2933, 2869, 2811, and 2479 cm^−1^ were found; in the spectrum of sample **3**, three bands at 2933, 2869, and 2811 cm^−1^; while, in the spectrum of the HKUST-1***mw*** sample, eight bands at 3094, 3061, 2997 (sh), 2933, 2895 (sh), 2811, 2664, and 2479 cm^−1^ were observed. The broad band with a maximum of about 3407 cm^−1^ corresponds to the stretching vibrations of the Si–OH groups linked by a weak hydrogen bond in the mesoporous silica matrix (BPS) [73,76].

The bands at 3086 and 3066 cm^−1^, which are clearly visible in a large scale of the HKUST-1@BPS (sample **3**) spectrum, practically coincide with the bands in the HKUST-1 ***mw*** spectrum (Appendix A).

Broad bands in the spectra of all the samples (Figure 7) may include bands characterizing the overtone of the C = O band at 1668–1691 cm^−1^ (Figure 8), as well as bands characterizing the stretching vibrations of the aromatic moiety of the btc linkers [69,70,77,78]. In the region of 3130 to 3070 cm^−1^, there are bands from the stretching vibrations of C–H in aromatic rings [72,79,80]. The bands at 2933 and 2867–2869 cm^−1^ (Figure 7) characterize asymmetric and symmetric stretching vibrations in –CH_3_ and –CH_2_ fragments [80,81].

The spectrum of the individual HKUST-1 ***mw*** sample (Figure 8) contains absorption bands at 1673, 1615, 1599, 1499 (sh), 1464, 1420, 1399, 1359, 1251, 1177, 1106, 1022, and 939 cm*^−^*^1^.

The spectrum of the **1** composite (Figure 8) contains absorption bands at 1687, 1673, 1594, 1499, 1450, 1420, 1390, 1251, 1106, 939, and 901 cm^−1^.

The spectrum of the composite HKUST-1@BPS sample **1** (black) (Figure 8) contains absorption bands at 1687, 1673, 1594, 1499, 1450, 1420, 1390, 1251, 1106, 939, and 901 cm^−1^. The spectrum of the composite HKUST-1@BPS sample **2** (red) contains absorption bands at 1673, 1587, 1450, 1420, 1394, 1359, 1332, 1251, 1106, 1022, 939, and 885 cm^−1^. The spectrum of the composite HKUST-1@BPS sample **3** (blue) contains absorption bands at 1691, 1654 (sh), 1587, 1499, 1450, 1420, 1390, 1251, and 1177 cm^−1^, which refer to the vibrations of the framework in these HKUST-1@BPS composite materials [69,70,72,76,77,78,79,80]. The same area includes absorption bands from bending vibrations of CH_3_, i.e., 1454 (antisym.) and 1394 (sym.) cm^−1^. The absorption band at 1251 cm^−1^ refers to antisymmetric stretching vibrations of the Si–O bond, which is characteristic of the BPS matrix [82,83,84,85,86,87,88,89,90].

### 3.6. The Hold-Up Volumes and the Volumes of a Stationary Phase for the HPLC Columns Filled with the HKUST-1@BPS Composite Materials, the Pristine HKUST-1, and Bare BPS Powders

In order to calculate the values of the retention factor and thermodynamic characteristics of adsorption and to estimate the porosity of the studied chromatographic columns, we determined the values of the column hold-up volume, *V*_M_, the volume of a stationary phase in a column, *V*_S_, and the phase ratio, *φ*. It should be noted that the knowledge of these parameters of chromatographic columns facilitates understanding the accessibility of the stationary phase pore structure towards the mobile phase molecules during the HPLC process. In addition to the textural characteristics determined by low-temperature nitrogen adsorption, these parameters enhance the characterization of chromatographic columns because they show, unlike the nitrogen adsorption method, actual free volume in the column accessible towards a mobile phase under HPLC conditions.

The column hold-up volume, also known as a column “dead” volume, can be considered as a measure of the free space in the chromatographic column, but it depends on the column dimensions. Therefore, in order to estimate the porosity of the columns and to compare the studied columns, the values of the phase ratio are more useful. In this work, the values of the hold-up volume were determined by utilizing the direct method of static pycnometry. Therefore, the measured hold-up volumes can be considered as intrinsic properties of the prepared chromatographic columns that do not include extra-column volumes. Moreover, the calculated values of the volume of a stationary phase in a column and, consequently, the values of the phase ratio are also intrinsic properties of the studied columns. Generally, the porosity of a stationary phase decreases, as is well known, with increasing the value of the phase ratio of a chromatographic column.

The minimum porosity is inherent for the bare BPS reference sample: *V*_S_ = 0.176 cm^3^, *V*_M_ = 0.319 cm^3^, *φ* = 0.552. On the contrary, the maximum porosity is characteristic of the pristine HKUST-1 reference sample: *V*_S_ = 0.204 cm^3^, *V*_M_ = 0.627 cm^3^, *φ* = 0.325. The columns filled with two composite materials have intermediate values of the phase ratio and, consequently, medium porosity in comparison with the columns with the pristine HKUST-1 and bare BPS: for the column filled with the **1** sample, *V*_S_ = 0.216 cm^3^, *V*_M_ = 0.412 cm^3^, *φ* = 0.524; for sample **3**, *V*_S_ = 0.112 cm^3^, *V*_M_ = 0.241 cm^3^, *φ* = 0.465. The column filled with composite **2** also exhibits the minimum phase ratio, which is the same as for the column with the pure HKUST-1 material: *V*_S_ = 0.129 cm^3^, *V*_M_ = 0.400 cm^3^, *φ* = 0.323.

As we can see from the comparison of the data presented here and in Table 1, the composite material **2** and the column with that exhibit the value of the phase ratio lowest amid the other composites, and, hence, have the highest porosity, as well as the maximum values of the specific surface area and the content of HKUST-1 in the matrix among the other composites. At the same time, the textural characteristics of composites **1** and **3**, such as the specific surface area and the HKUST-1 content in a matrix, as well as the values of porosity, are lower and closer to the textural characteristics of the bare BPS sample, while the phase ratios are higher for them (Table 1).

Expressly, according to the described data, there is a strong relationship between the characteristics of the prepared chromatographic columns and the textural features of the porous materials used as the stationary phases. Furthermore, it was shown that the prepared chromatographic columns have a large free volume accessible for the mobile phase molecules during the HPLC process. Therefore, according to the demonstrated regularities, the preparation conditions for the composite materials impact the values of the phase ratio and the free porous space in the chromatographic columns. Thus, they can affect the properties and operation of the columns upon HPLC.

### 3.7. The Influence of the Mobile Phase Flow Rate on the Benzene Retention upon HPLC on the HKUST-1@BPS Composite Materials

In order to study the accessibility of the micropores of the HKUST-1@BPS materials under conditions of HPLC towards molecules of an adsorbate in comparison with the pristine HKUST-1 and bare BPS samples, the benzene retention dependences on the mobile phase flow rate were measured. As we can see in Figure 9, for the column filled with the pristine HKUST-1 powder, the values of the benzene retention factor increase with decreasing the mobile phase flow rate across the entire region of variation of the eluent velocity. It reflects no adsorption equilibrium during HPLC on this stationary phase, even at the low values of the mobile phase flow rate. Probably, the observed pattern is induced because of the abundant presence of micropores in the HKUST-1 structure (Table 1) and, consequently, by diffusion difficulties during the adsorbate mass transfer in these pores upon the dynamic process of HPLC. Recently [56], we suggested that the adsorbate molecules cannot penetrate into HKUST-1 micropores under the high mobile phase flow rates because of diffusion limitations, while they can do under the low mobile phase flow rates. Therefore, no adsorption equilibrium during the pristine HKUST-1 powder exploitation as a stationary phase in HPLC, in our opinion, signifies the necessity of improving the surface chemistry and, especially, chromatographic properties of this matrix through enhancing its porous structure.

The almost consummate equilibration appears across the whole range of the mobile phase flow rates when utilizing the bare BPS powder as the stationary phase upon liquid-phase adsorption under conditions of HPLC (Figure 9). However, the bare BPS powder could exhibit poor adsorption selectivity during separating by HPLC because of the absence of required adsorption sites existing in the HKUST-1 framework [59]. Meanwhile, this material has enhanced mass transfer properties due to the presence of a large amount of mesopores (Table 1). Therefore, it makes sense to utilize the BPS porous structure as a carrier for the HKUST-1 matrix to create stationary phases manifesting simultaneously satisfactory mass transfer properties and, hence, equilibration, as well as sufficient adsorption selectivity during separation using HPLC.

As represented in Figure 9, the prepared chromatographic columns filled with the created HKUST-1@BPS composite materials show a close to perfect equilibration across the whole range of the mobile phase flow rates because the retention factor value does not depend significantly on the flow rate of the eluent. The appearance of the plateau on the dependences of the retention factor on the eluent flow rate in the case of materials **1**, **2,** and **3** persuades us that there are no diffusion limitations, which could restrict the HPLC process utilizing the studied composites. Actually, unlike the determined dependence for the column with the pure HKUST-1 sample, the patterns presented in Figure 9 for the composites are closer and similar to the dependence that is characteristic of the column filled with the bare BPS powder. Despite the presence of HKUST-1 micropores in these composite adsorbents, the simultaneous availability of the transport mesopores of the BPS carrier probably assists the mass transfer of the adsorbate throughout the porous layer inside the chromatographic columns. Probably, in the case of applying the composites as stationary phases, no obstacles constrain the penetration of adsorbate molecules into HKUST-1 micropores. Consequently, it permits to reach the adsorption equilibrium not only utilizing the low eluent velocities but even upon operation at the middle flow rates of the mobile phase. The observed behavior of the investigated chromatographic columns filled with the created composite materials is the prosperity for their possible application as stationary phases for the operation under HPLC conditions and upon other dynamic liquid-phase adsorption methods, such as liquid-phase extraction, purification, etc.

It can be seen from the comparison of the data presented in Table 1 and Figure 9 that there is a strong relationship between the micropore specific volume of the studied composite materials and their ability to adsorb benzene molecules from the liquid phase as the stationary phases under the conditions of HPLC. In fact, the values of the retention factor are maximum across the whole region of variation of the eluent flow rates for the column filled with the bare BPS powder compared to the composite materials (Figure 9). For the columns filled with the composites, the retention factor values are lower than for the column with the bare BPS powder and decrease in the following order across the entire range of variation of the eluent velocity: **1**, **2**, **3**. The same order is observed for diminishing values of the micropores’ specific volumes of the investigated composite materials (Table 1). Moreover, the size of embedded HKUST-1 crystallites in the HKUST-1@BPS composite structure increases in the identical sequence (Figure 2). Embedding the MOF phase into mesopores of the BPS matrix and blockage of these pores are possible accounts for the lower benzene retention onto the hybrid adsorbents compared to the bare BPS material, while decreasing the microporous space in the matrix amid the composites also probably attenuates the ability of benzene molecules to retain upon HPLC while utilizing the hybrid stationary phases. The benzene retention factor on the column filled with the pristine HKUST-1 powder, as previously mentioned, is not constant and depends on the flow rate of the mobile phase. Hence, the retention factor values are the highest among the investigated materials at the reduced flow rates and lowest at the fast flow rates (Figure 9). It reflects a pronounced relationship between the accessibility of HKUST-1 micropores towards adsorbate molecules and their chromatographic retention upon HPLC utilizing HKUST-1 contained stationary phases. Expressly, the fast mobile phase flow rates tarry penetrating adsorbate molecules into HKUST-1 micropores.

Therefore, according to the performed analysis of the dependences presented in Figure 9, the prepared HKUST-1@BPS composite materials can function as the stationary phases for the HPLC columns, unlike the pristine HKUST-1 powder, due to the synergetic effect of both components. It was shown that the HKUST-1 and BPS components cooperating in the structure of the resulting materials allow outflanking restrictions discouraging utilizing the pristine MOF and the bare biporous silica as stationary phases in HPLC. The pivotal role of the possibility of the penetration of the adsorbate molecules into HKUST-1 micropores and the accessibility of BPS transport mesopores during liquid-phase adsorption upon HPLC were revealed. At the same time, preparation conditions for the composite materials directly influence their adsorption capability, which is reflected in the values of the benzene retention factor upon HPLC.

### 3.8. The Impact of the Mobile Phase Composition on the Benzene Retention in HPLC on the HKUST-1@BPS Composite Materials

In order to reveal the intermolecular interactions between the adsorbate and active sites in the HKUST-1@BPS porous structure when varying the composition of the mobile phase, the dependences of the retention factor on the isopropanol volume % in the mixed eluent were determined. Investigation of these dependences presumably will expand the understanding of the chromatographic mode that occurred in a studied case.

In the initial region of the measured dependences (Figure 10), for the composite materials **1** and **2** with the highest values of the micropore specific volume (Table 1), there is a narrow interval where the retention factor instantaneously reduces when increasing the isopropanol volume fraction in the mobile phase. Competitive adsorption and binding of isopropanol molecules with the specific Cu^2+^ adsorption sites in the HKUST-1 micropores and subsequent blockage of these centers towards adsorbate (benzene) molecules are possible accounts for the appearance of the observed pattern (Figure 10).

Subsequently, for the composites **1** and **2**, after a short region of the initial decrease, there is an acute increase in the retention factor values (Figure 10). To explain this phenomenon, we have to consider in Section 3.10 the influence of the eluent composition on the thermodynamic characteristics of adsorption onto the studied materials.

It should be noticed that the most dramatic changes, i.e., initial reducing and subsequent growth, of the benzene retention, when varying the eluent composition, as is seen in Figure 10, are characteristic of the behavior of the most homogeneous composite material **1**. For material **3,** with inhomogeneous morphology rather similar to a mechanical mixture of components, on the contrary, there is a different pattern with no initial decrease in the retention factor with an increase in the isopropanol volume fraction in the mobile phase. Contrary to the column with sample **1**, the retention factor values in the initial region for sample **3** immediately and slowly increase, and then gradually diminish at higher contents of isopropanol in the liquid phase.

Thus, for composites **1** and **2**, there is a region in the medium range of the variation of the mobile phase composition that is absent for composite **3**. After this interruption in decreasing the values of the retention factor with the increase in the isopropanol volume fraction, the retention factor when utilizing materials **1** or **2** again starts to diminish to the complete loss of the possibility of the column to retain adsorbate molecules at the higher isopropanol contents in the eluent (Figure 10). Obviously, such a reduction in the retention factor values with the increase in the isopropanol content in the eluent can be explained by the solvation of adsorbate molecules by isopropanol in the liquid phase.

Therefore, in general, the well-known quasi-normal-phase mode of HPLC manifests when the composite materials **1** or **2** are utilized as stationary phases in combination with weakly polar mobile phases at the initial and high concentrations of isopropanol in the mobile phase, excluding the short region of increasing the retention factor at the medium eluent compositions (Figure 10). Overall, for composite **3**, the quasi-normal-phase variant of HPLC occurs across the whole region of variation of the mobile phase composition (Figure 10). This mode was previously described for HPLC using hyper-cross-linked polystyrenes and porous graphitic carbon [2]. The π–π adsorbate–adsorbent binding prevails the intermolecular interactions of adsorbate molecules (solvation) with the mobile phase components during this HPLC mode. Still, this binding is not so strong as during normal-phase HPLC when the pure silica and *n*-hexane are used as a stationary and mobile phase, respectively [2].

The HKUST-1 and BPS reference samples during their operation as stationary phases in the HPLC columns at varying eluent compositions exhibit the dependences of the retention factor versus the isopropanol volume fraction that are similar to that of the composite material **3**, which is a mechanical mixture of them.

Therefore, the chemical nature, porous structure, and preparation conditions of the studied novel HKUST-1@BPS composite materials and the pristine HKUST-1 and bare BPS samples have a pronounced effect on the chromatographic mode manifesting during their operation as the stationary phases for HPLC. It was revealed that, under the varying mobile phase composition, the columns filled with the **1** and **2** composites exhibit the chromatographic behavior evidencing the synergetic effect of the MOF and BPS components. In contrast, the HPLC properties of the column filled with the **3** powder indicate the separate working of both initial components because this column has a pattern closer to the columns filled with the pure components. These facts highlight the possibility of utilizing the HPLC technique to investigate the surface properties of novel composite materials.

### 3.9. The Effect of the Chromatographic Column Temperature on the Benzene Liquid-Phase Adsorption onto the HKUST-1@BPS Composite Materials

Besides the mobile phase composition, the chromatographic column temperature also affects the values of the benzene retention factor upon HPLC utilizing the studied novel HKUST-1@BPS composite materials as stationary phases. It was revealed that the benzene liquid-phase adsorption onto the bare BPS material from the *n*-hexane–isopropanol mobile phases is always an exothermic process, so increasing the temperature induces decreasing the chromatographic retention not depending on the eluent composition. On the contrary, during HPLC using the pristine HKUST-1, the pattern of the temperature dependence of the retention is determined by the isopropanol content in the eluent. Thus, the benzene liquid-phase adsorption is exothermic when utilizing the mobile phases based on the pure *n*-hexane or *n*-hexane–isopropanol mixtures with a polar component content lower than 0.4 vol.%. In contrast, for the eluent compositions with the higher isopropanol content or vice versa, endothermic adsorption is observed.

For the studied composite materials, the same as for the pristine HKUST-1, there is a pattern including substituting the exothermic character of adsorption by endothermic when changing the eluent composition. Figure 11 shows clearly that, in the case of the composite material **3,** depending on the eluent composition, the value of the distribution constant reflecting the energy of adsorbate–adsorbent interacting decreases or increases with the rise in the chromatographic column temperature. When the benzene liquid-phase adsorption occurs from the media of pure *n*-hexane or *n*-hexane–isopropanol mixtures with the content of a polar component not higher than 0.3 vol.%, increasing the column temperature results in decreasing the values of the distribution constant. On the contrary, during the benzene adsorption from binary *n*-hexane–isopropanol solutions with the alcohol content 0.4 vol.% and higher, increasing the column temperature causes increasing chromatographic retention. Consequently, in the initial region of increasing the isopropanol content in the eluent, we observed exothermic adsorption, while subsequently increasing the isopropanol volume fraction induces transforming the adsorption pattern to endothermic (Figure 11).

Probably, the change in an adsorption mechanism, and, particularly, the transforming of the adsorption phase, is an account for changing the pattern of the temperature dependence of the distribution constant. To justify this hypothesis, we have to discuss the peculiarities of liquid-phase adsorption thermodynamics onto the HKUST-1@BPS composite materials in the following Section 3.10. Herein, it should be noticed that the measured dependences of the logarithm of distribution constant versus inverse temperature are linear to calculate the thermodynamic characteristics of adsorption (Figure 11). The crossing in the same point of the adsorption isosteres in Figure 11 reflects a compensation effect that confirms the close mechanism of the benzene adsorption from the mobile phases with the isopropanol content varying from 0.4 to 5 vol.%. For the same reason, a close mechanism is also characteristic of the benzene adsorption from the mobile phases with the isopropanol content 0.2 and 0.3 vol.%. In contrast, benzene adsorption processes from pure *n*-hexane and *n*-hexane–isopropanol with minimum isopropanol content occur following other mechanisms (Figure 11).

It was shown that the chromatographic column temperature has a pronounced effect on the liquid-phase adsorption on the prepared novel HKUST-1@BPS composite materials upon HPLC. Furthermore, besides the case of the bare BPS material, the pattern of the temperature dependence is determined by the solvent composition. It was revealed that changing the pattern of the temperature dependence of the chromatographic retention de-pending on the mobile phase composition is a feature of HKUST-1-contained adsorbents. Moreover, the HKUST-1 micropores definitely contribute to converting exothermic adsorption by endothermic when varying the eluent composition. The primary hypotheses formulated based on the previous HPLC examinations can be confirmed utilizing the obtained thermodynamic data analysis.

### 3.10. Thermodynamics of Benzene Liquid-Phase Adsorption onto the HKUST-1@BPS Composite Materials upon HPLC

In order to apprehend the energetics of the liquid-phase adsorption onto the HKUST-1@BPS samples under varying compositions of the bulk solution from which liquid-phase adsorption occurs, we have measured the dependences of the thermodynamic characteristics of benzene adsorption on the isopropanol volume % in the eluent under HPLC conditions (Figure 12). The obtained medium–property relationships, in our opinion, facilitate comprehension of the adsorption mechanisms upon HPLC utilizing the prepared composite materials as stationary phases.

As seen in Figure 12, for the studied HKUST-1@BPS composites, at the initial isopropanol concentrations in the eluent (0.1–0.3 vol.%), the energy of the adsorbate–adsorbent interaction and localization of benzene molecules in pores are rather high. Still, by increasing the isopropanol content up to 0.4 to 1.0 vol.%, the thermodynamic characteristics of adsorption gradually reduce in absolute values and even have positive values.

Moreover, in the range of the isopropanol content of 0.2 to 0.4 vol.% for all the composites, there is a dramatic change in the enthalpy and entropy values, reflecting restructuring the adsorbed layer (Figure 12). Therefore, for composites **1** and **2,** there are the maximum negative surges of enthalpy and entropy when increasing the isopropanol content in the mobile phase (Figure 12a,b). Meanwhile, for composite **3**, these negative surges are lower in absolute values, and the maximum equivalent positive upsurges of enthalpy and entropy are observed (Figure 12c). Such a complicated pattern of the obtained dependences probably reflects the change in the mechanism of benzene liquid-phase adsorption onto the composite materials upon varying mobile phase composition, which is induced by restructuring the adsorbed phase.

For all the studied composite materials, a subsequent increase in the isopropanol content in the mobile phase induces diminishing the absolute values of enthalpy, entropy, and Gibbs energy (Figure 12). This pattern is observed because replacing pre-adsorbed isopropanol with benzene molecules under conditions of competitive adsorption requires additional energetic consumptions and induces the release of additional degrees of freedom. It means that, firstly, energetic consumptions for replacing pre-adsorbed isopropanol molecules and intensifying solvation of benzene molecules by isopropanol and, secondly, the release of additional degrees of freedom upon pre-adsorbed isopropanol desorption into the bulk solution of the eluent increase with isopropanol content increasing in the liquid phase.

It should be noted that these dramatic changes of the values of enthalpy and entropy, in fact, are not reflected in the dependences of the Gibbs energy on the solvent composition (Figure 12). Possibly, the mutual compensation of enthalpy and entropy is an account for the mentioned phenomenon. First, during benzene adsorption from pure *n*-hexane, the values of the Gibbs energy for all composites are rather high. Upon subsequent increasing the isopropanol volume fraction in the mobile phase, the Gibbs energy initially decreases in absolute value because of possible blockage of open metal Cu^2+^ adsorption sites in HKUST-1 micropores by isopropanol molecules. Then the energy of adsorption slowly increases in an untrivial way due to a manifestation of peculiar mechanism of liquid-phase adsorption on microporous MOF that previously has been discovered for MIL-53(Al) framework [91]. In this area, there is even an entropy-driven mechanism of liquid-phase adsorption at the higher isopropanol volume fractions in the eluent (Figure 12). The subsequent diminishing strength of π–π adsorbate–adsorbate interaction can be explained by intensive solvation of benzene by isopropanol molecules (Figure 12). In this final range of the varying isopropanol content in the eluent, the chromatographic systems based on the HKUST-1@BPS materials exhibit behavior intrinsic to the so-called quasi-normal-phase mode of HPLC [2].

In the initial region of varying the isopropanol content in the mobile phase, there is a maximum difference between the values of Gibbs energy measured at the same eluent composition for the adsorption on the different composite materials (Figure 12). Subsequently, the studied materials exhibit, in fact, equal adsorption activity because of intensive adsorbate solvation by isopropanol. In the initial range of isopropanol concentrations, **1** and **2** materials have higher adsorption activity even than the pure HKUST-1 stationary phase (Figure 12). Composite **3** exhibits behavior closer to that of the pristine HKUST-1 material, probably because this composite material is similar to a mechanical mixture of pure HKUST-1 and BPS components.

To estimate separately the contributions of the MOF and silicate phases to an implemented retention mechanism on the studied composites, we should compare the dependencies represented in Figure 12 with similar dependences that are intrinsic for the pure HKUST-1 and BPS materials (Figure 13). For the columns filled with the pristine HKUST-1 (Figure 13a) and bare BPS (Figure 13b) powders, there are generally similar patterns of the dependences of the thermodynamic characteristics of adsorption on the eluent composition. However, the pattern obtained for the pristine HKUST-1 material is closer to patterns observed for the studied composites. It means that, during liquid-phase adsorption upon HPLC utilizing the prepared composite materials, chromatographic retention is implemented primarily due to adsorption inside the HKUST-1 micropores. At the same time, the BPS mesopores serve as transport pores to mass transfer throughout the chromatographic column.

Therefore, the strong impact of the mobile phase composition on the thermodynamic characteristics of adsorption explains the previously described crucial influence of the solvent on separation behavior [92]. Furthermore, the implemented analysis of the thermodynamics of adsorption confirms and develops the assumptions performed during the study of the effect of the mobile phase composition on the chromatographic retention when utilizing the columns with the prepared stationary phases.

## 4. Conclusions

In this work, the novel HKUST-1@BPS composite materials were synthesized using the microporous HKUST-1 metal-organic framework and BPS with a bimodal mesopore size distribution as components through the original MW-assisted procedure at atmospheric pressure. For this purpose, a one-step in situ approach was utilized. Both BPS loading and synthesis mode were rationally varied in the composite preparation process. It was found that these parameters have a pronounced impact on their physicochemical characteristics, such as structural, compositional, textural, and morphological parameters. The most homogeneous **1** composite system with a high dispersion of small (20–30 nm) HKUST-1 nanocrystallites was obtained using the preliminary grinding together of all the reaction components and moderate (3.0 g) BPS loading. The **3** composite synthesized using the separate H_3_(btc) and Cu^2+^ solutions and largest BPS loading has the biggest HKUST-1 crystallites and inhomogeneous morphology rather similar to a mechanical mixture of components.

The produced HKUST-1@BPS composites can function as stationary phases for HPLC, unlike pristine HKUST-1 material, due to the synergetic effect of both components based on the preliminary enhanced mass transfer throughout the silica mesopores and the subsequent adsorption into the HKUST-1 micropores. It was established that penetrating adsorbate molecules into the micropores of the pure HKUST-1 framework solely realizes at the extremely low flow rates of the mobile phase. It means that, upon the HPLC process using the pristine HKUST-1 material as a stationary phase, adsorption equilibrium can be achieved only under conditions of micro- and nano-flow HPLC. Another way of utilizing pure HKUST-1 is static liquid-phase adsorption techniques, e.g., solid-phase extraction or deep purification of substances. Upon HPLC using the bare BPS matrix as a stationary phase, adsorption equilibrium can be reached even at the eluent middle flow rates, but this material does not have unique selective properties. Due to the presence of transport mesopores in the BPS structure, creating MOF-containing composite materials based on the BPS carrier enhances the operation of the resulted materials as stationary phases for HPLC. Therefore, the exploitation of the HKUST-1@BPS composites overcomes restrictions manifested during the utilization of separate components and provides higher efficiency than that revealed by the precursors.

The suggested working mechanism involves the initial deactivation of the open metal Cu^2+^ adsorption centers of the HKUST-1@BPS composites upon the increase in the polar component content in the mobile phase, the subsequent unique genesis of new active sites at the medium range of the isopropanol content variation in the eluent, and then the behavior of the studied chromatographic systems, which is similar to the quasi-normal-phase HPLC pattern, while further increasing the isopropanol volume fraction in the eluent.

According to the obtained thermodynamic data, liquid-phase adsorption into HKUST-1 micropores at the medium isopropanol concentration in the eluent follows the unique entropy-driven mechanism previously described for the MIL-53(Al) framework. It includes the transition of adsorbate molecules during adsorption from the unordered bulk solution of the mobile phase to a more structured solvent in the micropores, which is ordered due to strong interactions with their walls. A decrease in the number of degrees of freedom during this transition comes down to positive values of entropy changes upon adsorption.

The preparation strategy used for creating the HKUST-1@BPS composites along with the BPS content in them also plays a dominating role in their adsorption performance, including the adsorption mechanism.

## Figures and Tables

**Figure 1 polymers-14-01373-f001:**
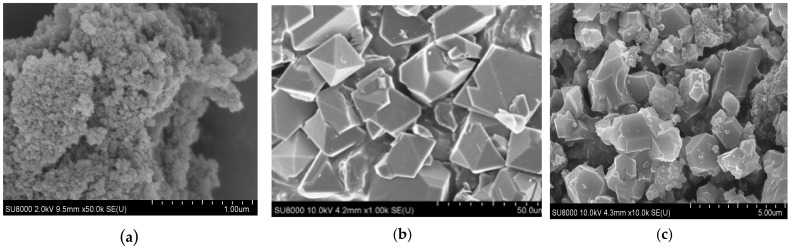
SEM micrographs of the bare BPS (**a**) and pristine HKUST-1 ***mw*** (**b**) components and HKUST-1@BPS (sample **1**) composite material (**c**).

**Figure 2 polymers-14-01373-f002:**
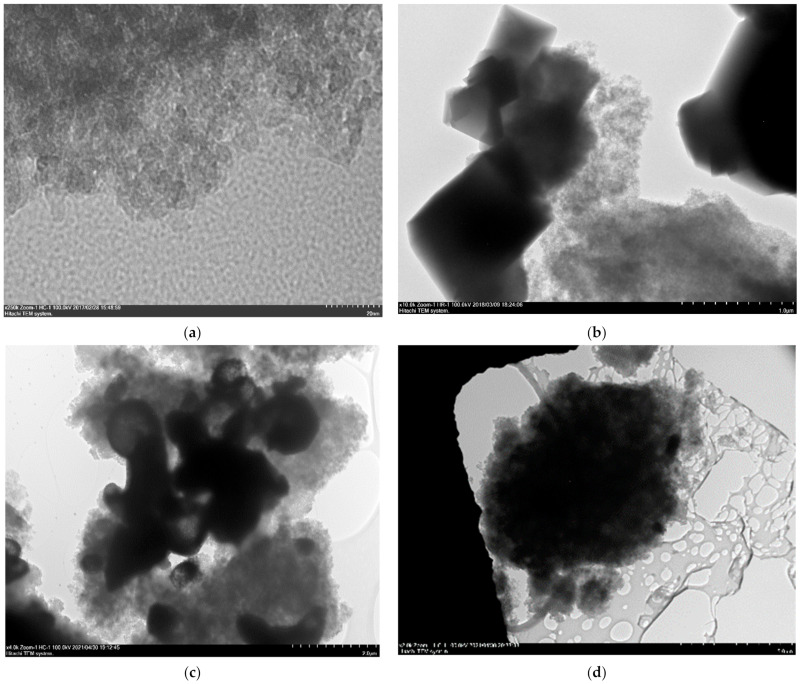
TEM micrographs of the bare BPS silica (**a**) and the HKUST-1@BPS composite samples **1** (**b**), **2** (**c**), and **3** (**d**). The dimensions of the scale bars on TEM images are equal to 20 nm (**a**), 1 μm (**b**), 2 μm (**c**), and 5 μm (**d**).

**Figure 3 polymers-14-01373-f003:**
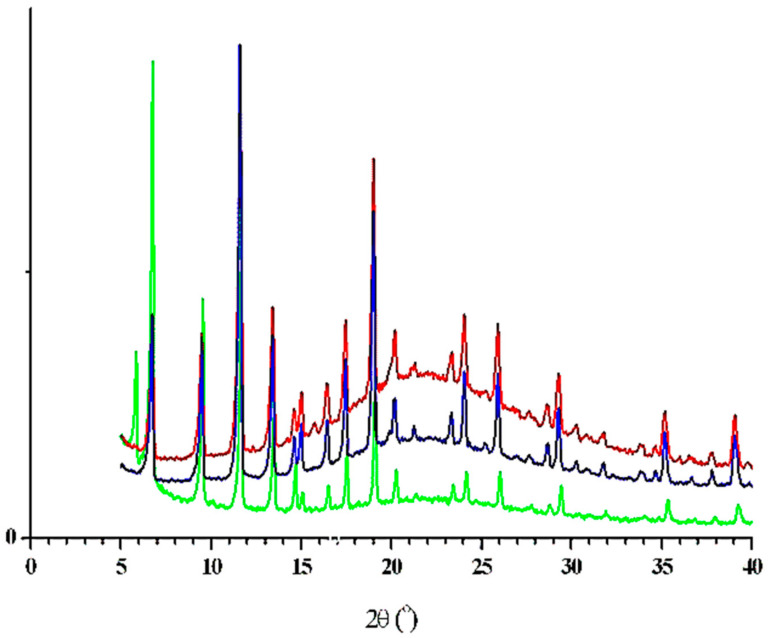
The combined presentation of the XRD patterns of the **1** (green), **2** (blue), and **3** (red) samples.

**Figure 4 polymers-14-01373-f004:**
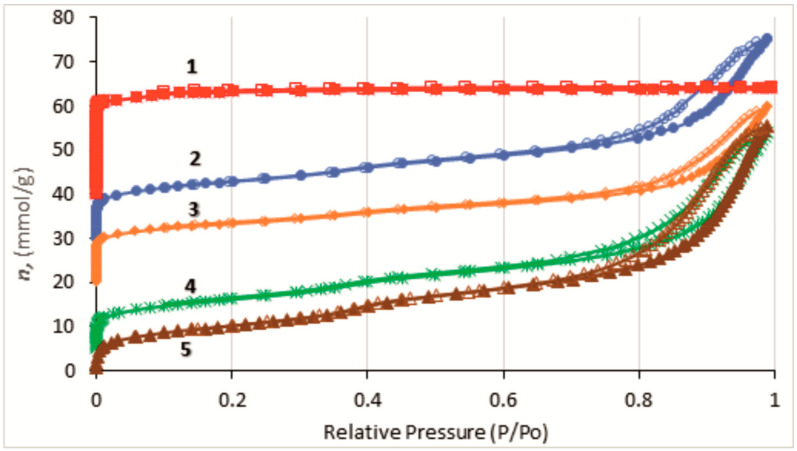
N_2_ adsorption isotherms at 77 K for the HKUST-1@BPS composites, the HKUST-1 ***mw*** reference sample, and the bare BPS matrix.

**Figure 5 polymers-14-01373-f005:**
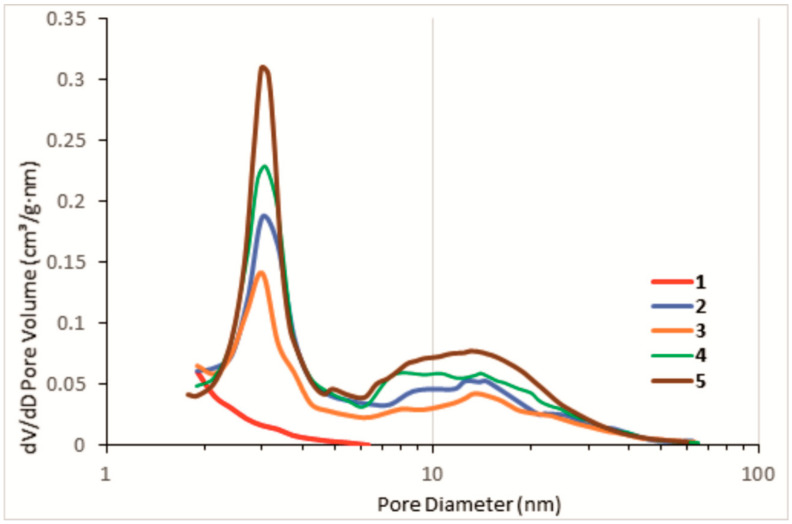
Mesopore size distribution for the HKUST-1@BPS composites, the pristine HKUST-1 ***mw*** reference sample, and the bare BPS matrix.

**Figure 6 polymers-14-01373-f006:**
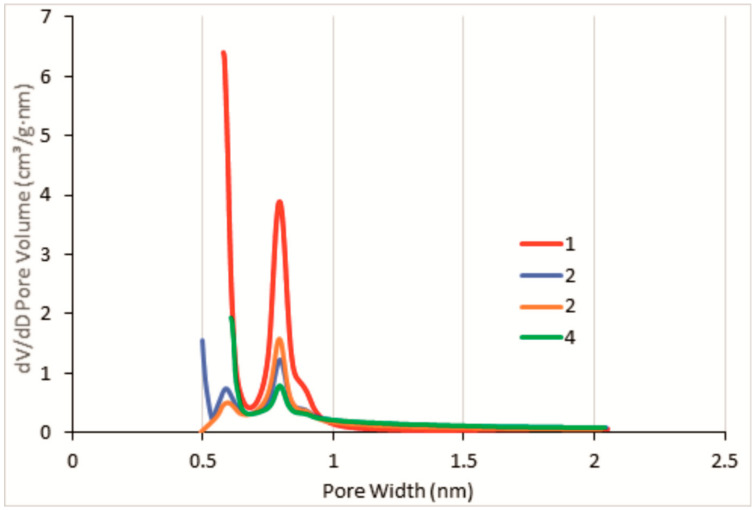
Micropore size distribution calculated by the Horvath–Kawazoe method for the HKUST-1@BPS composites and HKUST-1 ***m******w*** reference sample.

**Figure 7 polymers-14-01373-f007:**
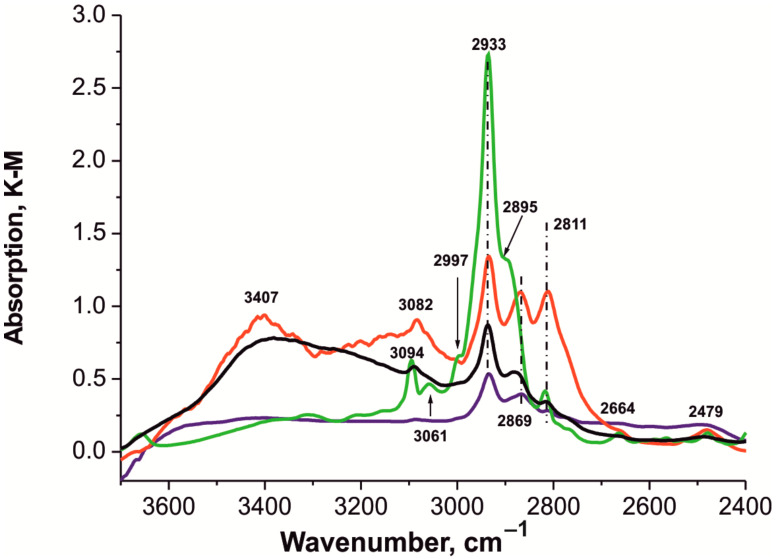
DRIFT spectra of the **1** (black), **2** (red), and **3** (blue) composites and the HKUST-1 ***mw*** reference sample (green) in the frequency range of 3700–2400 cm^−^^1^.

**Figure 8 polymers-14-01373-f008:**
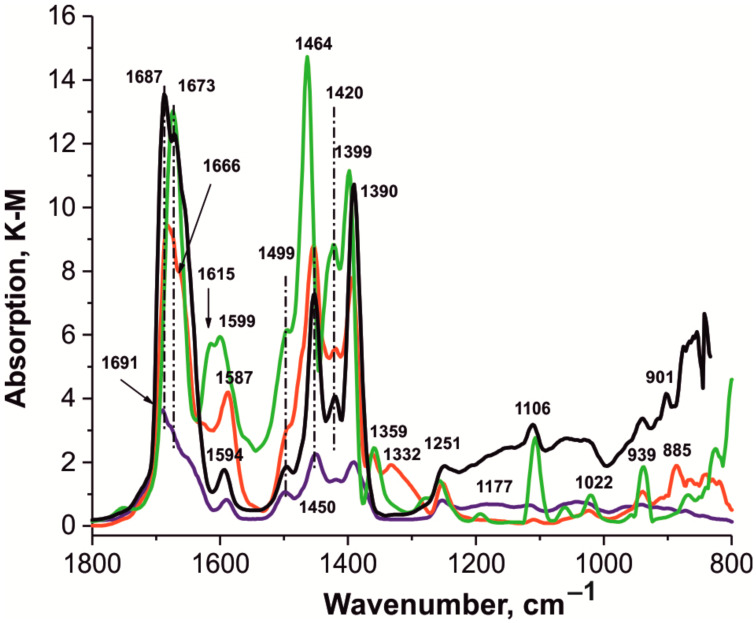
DRIFT spectra of the ***1*** (black), **2** (red), and **3** (blue) HKUST-1@BPS composites and the HKUST-1 ***mw*** reference sample (green) in the frequency range of 1800–750 cm^−1^.

**Figure 9 polymers-14-01373-f009:**
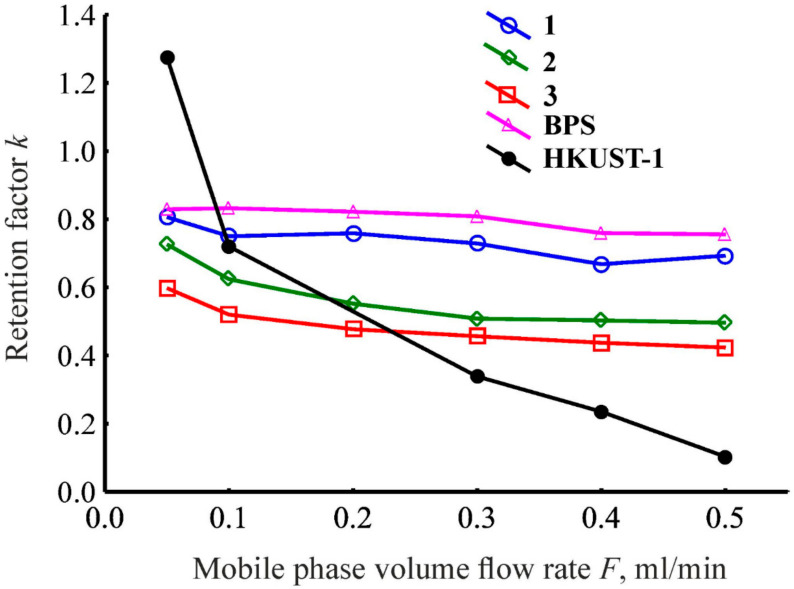
Dependences of the benzene retention factor on the volumetric flow rate of the mobile phase for the HKUST-1@BPS composites, the pristine HKUST-1, and bare BPS samples.

**Figure 10 polymers-14-01373-f010:**
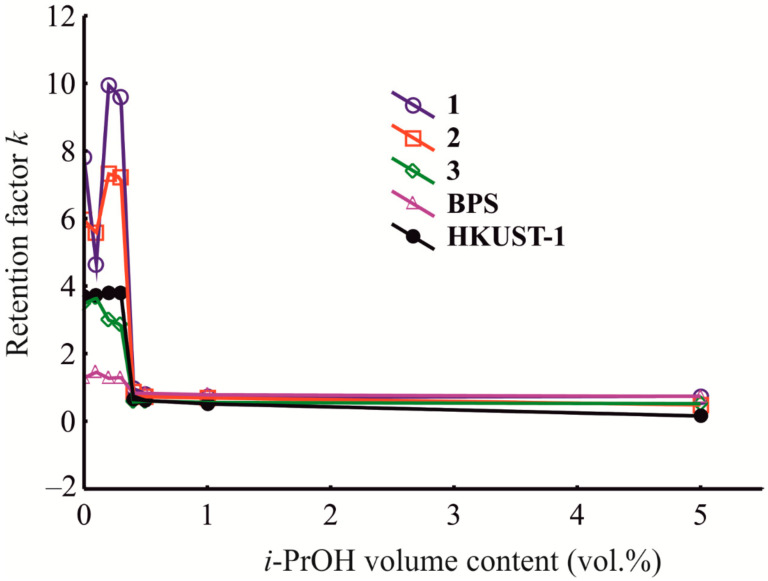
Dependences of the benzene retention factor on the *i*-PrOH content (vol.%) in the mobile phase for the HKUST-1@BPS composites, pristine HKUST-1, and bare BPS samples.

**Figure 11 polymers-14-01373-f011:**
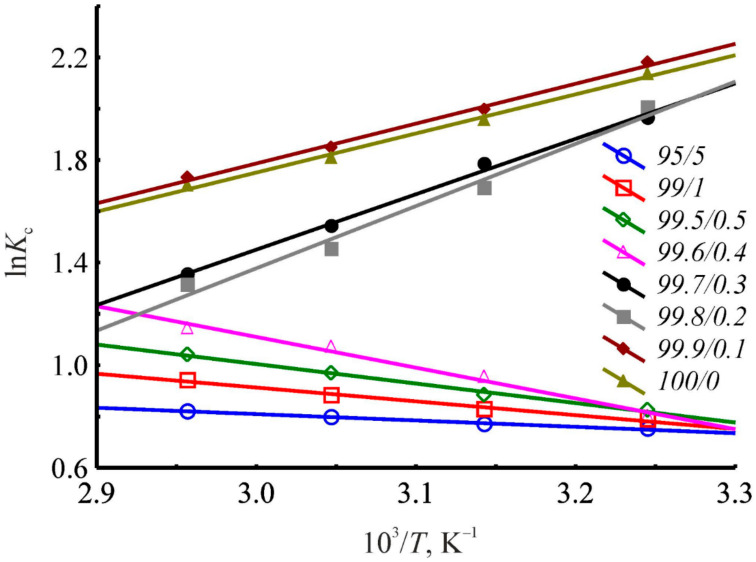
Dependences of the logarithm of the benzene distribution constant on the reciprocal temperature during adsorption onto the HKUST-1@BPS (**3**) composite.

**Figure 12 polymers-14-01373-f012:**
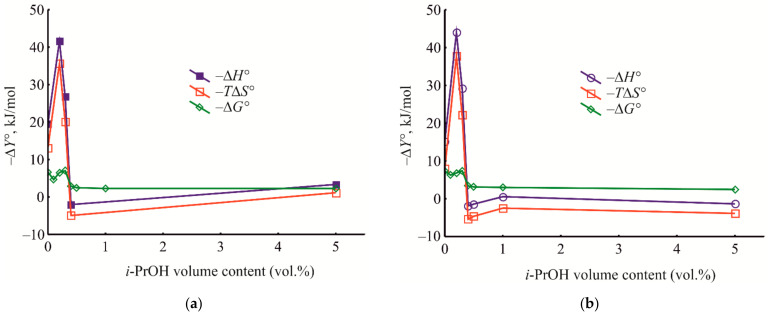
Dependences of the thermodynamic characteristics of benzene adsorption onto the HKUST-1@BPS composites **1**, **2**, **3** (**a**–**c**) on the *i*-PrOH content (vol.%) in the mobile phase.

**Figure 13 polymers-14-01373-f013:**
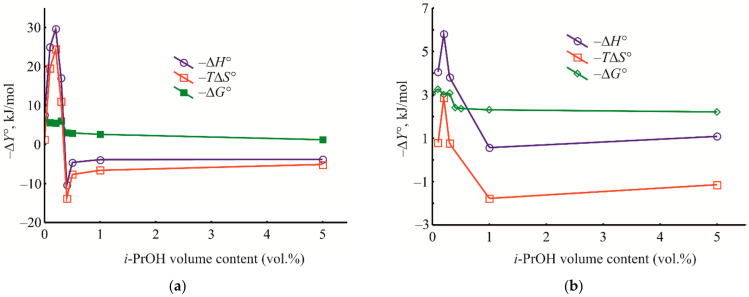
Dependences of the thermodynamic characteristics of benzene adsorption onto the pristine HKUST-1 (**a**) and bare BPS (**b**) samples on the *i*-PrOH content (vol.%) in the mobile phase.

**Table 1 polymers-14-01373-t001:** Textural characteristics of the HKUST-1@BPS composite materials and the HKUST-1 ***mw*** reference sample.

Sample No on Figure 4, Figure 5 and Figure 6	Material	A_BET_ *, m^2^/g	V_total,_ ^a^ cm^3^/g	V_micro,_ ^b^ cm^3^/g	V_meso,_ ^c^ cm^3^/g	HKUST-1, wt.%, **	HKUST-1, wt.%, ***	HKUST-1, wt.%, ****
1	HKUST-1 ***mw***	2049	0.836	0.790	0.046	100	100	100
2	**1**	962	1.569	0.184	1.385	28.4	25.3	23.3
	**1** Press 50 atm	942	1.336	0.396	0.940	–	–	–
3	**2**	1053	1.384	0.270	1.114	37.3	30.6	34.2
4	**3**	910	1.697	0.097	1.600	16.6	11.6	12.3
5	BPS	833	1.921	–	1.921	0	0	0

* ISO 9277. ** Determined by the BPS amount introduced in the synthesis. *** According to elemental analysis data. **** V_micro_/V_micro HKUST-1 *mw*_. ^a^ V_total_ was estimated from the adsorption value at *p*/*p*_0_ = 0.99; ^b^
*V_μ_* = *V_Σ_* − *V_meso_*. ^c^ The cumulative mesopore volume was calculated at 2 nm from the desorption branch of the isotherm by the BJH method and the standard thickness of the adsorption film.

## Data Availability

The data presented in this study are available on request from the corresponding authors.

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
