# Peer review of "Understanding the Working Mechanism of the Novel HKUST-1@BPS Composite Materials as Stationary Phases for Liquid Chromatography"

_polymers, 2022, doi:10.3390/polym14071373_

Round 1
Reviewer 1 Report
The authors have carefully and thoroughly addressed most of my queries/comments/suggestions. I recommend this manuscript be accepted for publication.
Reviewer 2 Report
Comment to authors as follows:
- The title of this manuscript is very long and authors should shorten it.
- The abstract is poor writing style and information and should include the significant numerical values obtained from the results presented in this paper.
- The definition and applications of composite materials should be presented in a new paragraph in the introduction section. In addition, some references are recommended to enhance the quality of this section.
https://journals.sagepub.com/doi/full/10.1177/26349833211007502
https://www.sciencedirect.com/science/article/abs/pii/S2452213918301608
https://iopscience.iop.org/article/10.1088/1757-899X/376/1/012074
- The applications and novelty of this study should report clearly in the last paragraph of the introduction.
- Thermal properties (TGA and DSC) are recommended to show the comparison in thermal stability of composites materials studied.
- The discussion of XRD and textural characteristics results are required to be explained with further details and compared with previous related literature.
Reviewer 3 Report
In the manuscript, the authors prepared HKUST-1@BPS composite materials and studied their applications in HPLC. The manuscript is well written and contains lots of information. I would recommend the acceptance of the manuscript after the following revision:
- An abbreviation should be defined as its first mention. For example, Microwave (MV)-assisted technique.
- In this work, three different composite systems were prepared. For sample 3, comparing to samples 1 and 2, it has different quantity of BPS silica and different preparation method. Since the two variables (the quantity of BPS and the preparation method) were changed at the same time in the system, how to determine which of the variables caused the observed differences between sample 3 and samples 1 and 2?
- Information about TEM study (sample preparation, accelerating voltage, instrument model, etc.) should be included in the characterization section.
- Clear scale bars should be included in SEM (Figure 1) and TEM (Figure 2) images.
- What are the SEM images of sample 2 and sample 3 composite materials look like?
- lines 436-440. The description is not clear. I suggest the authors to label the two fractions of HKUST-1 crystallites in the TEM image.
- Figure 3, there seems to be an extra peak for sample 1 at around 6 degree, what does this peak represent?
- Table1, sample5, is the Vmeso of 833 correct?
- Table 1, sample 1 press 50 atm, what is the surface area value after compressing?
- Line 540-550, the authors discussed in detail about the different peaks observed for the three composite samples. However, why those differences were observed was lack of discussion.
- some of the relevant references may be cited: synthesis and application of MOF material (Communications Chemistry 4.1 (2021): 1-10.; Chemistry–A European Journal 26, no. 61 (2020): 13788-13791; Journal of Chromatography A, 1363, 11-26.; Chemical Engineering Science, 124, 179-187.); MOF composites for HPLC (Microporous and Mesoporous Materials 263 (2018): 268-274.)
Round 2
Reviewer 3 Report
The authors have addressed my concerns, and I recommend the publication of this manuscript in the present form.
This manuscript is a resubmission of an earlier submission. The following is a list of the peer review reports and author responses from that submission.
Round 1
Reviewer 1 Report
Comments on polymers-1507350
This paper describes the synthesis of novel HKUST-1@BPS composite materials by using the microporous HKUST-1 metal-organic framework and BPS with a bimodal mesopore size distribution as components through the original MW-assisted procedure at atmospheric pressure. I recommend the publication of this manuscript in Polymers after minor revisions.
- The sections of “Abstract” and “Conclusions” are too long and should be rewritten for clarity.
- Whether the experiment at 3.1 has carried out parallel experiments to verify that the steps of the experiment will directly affect the structure of the product.
- The picture on line 432 is misquoted.
- In figure 3, what is the cause of the peak value between 5 and 6 degrees in the X-ray diffraction pattern of sample 1? I hope you can explain it.
- Line 480 refers to " On the contrary, the pressing increases the micropore volume for the 1 composite…". Does it mean that the structure is unstable?
- As mentioned in line 647 of the article, "there is no obstacle to the permeation of adsorbed molecules into HKUST-1 micropores." Is it conjecture or supported by experimental data?
- I would like to ask why the DRIFT spectra at Section 3.5 did not test and analyze the structure of sample 1.